# SynLogic: Synthesizing Verifiable Reasoning Data at Scale for Learning Logical Reasoning and Beyond

Junteng Liu[1,2], Yuanxiang Fan[2], Zhuo Jiang[2], Han Ding[2], Yongyi Hu[2], Chi Zhang[2],
Yiqi Shi[2], Shitong Weng[2], Aili Chen[2], Shiqi Chen[3], Yunan Huang[2], Mozhi Zhang[2],
Pengyu Zhao[2], Junjie Yan[2], Junxian He[1]

[1]The Hong Kong University of Science and Technology    [2]MiniMax
[3]The City University of Hong Kong
{jliugi, junxianh}@cse.ust.hk

## Abstract

Recent advances such as OpenAI-o1 and DeepSeek R1 have demonstrated the potential of Reinforcement Learning (RL) to enhance reasoning abilities in Large Language Models (LLMs). While open-source replication efforts have primarily focused on mathematical and coding domains, methods and resources for developing general reasoning capabilities remain underexplored. This gap is partly due to the challenge of collecting diverse and verifiable reasoning data suitable for RL. We hypothesize that logical reasoning is critical for developing general reasoning capabilities, as logic forms a fundamental building block of reasoning. In this work, we present SYNLOGIC, a data synthesis framework and dataset that generates diverse logical reasoning data at scale, encompassing 35 diverse logical reasoning tasks. The SYNLOGIC approach enables controlled synthesis of data with adjustable difficulty and quantity. Importantly, all examples can be verified by simple rules, making them ideally suited for RL with verifiable rewards. In our experiments, we validate the effectiveness of RL training on the SYNLOGIC dataset based on 7B and 32B models. SYNLOGIC leads to state-of-the-art logical reasoning performance among open-source datasets, surpassing DeepSeek-R1-Distill-Qwen-32B by 6 points on BBEH. Furthermore, mixing SYNLOGIC data with mathematical and coding tasks improves the training efficiency of these domains and significantly enhances reasoning generalization. Notably, our mixed training model outperforms DeepSeek-R1-Zero-Qwen-32B across multiple benchmarks. These findings position SYNLOGIC as a valuable resource for advancing the broader reasoning capabilities of LLMs. We open-source both the data synthesis pipeline and the SYNLOGIC dataset in `https://github.com/MiniMax-AI/SynLogic`.

## 1   Introduction

The success of Deepseek R1 [DeepSeek-AI et al., 2025] and OpenAI-o1 [Jaech et al., 2024] demonstrates the great potential of post-training in advancing strong reasoning capabilities. These works reveal that the core methodology behind these advancements is reinforcement learning with verifiable rewards (RLVR), inspiring numerous replication efforts focused on RL training. However, most of these works have concentrated on the mathematics and coding domains, primarily because it is straightforward to design binary reward rules in these areas [Zeng et al., 2025b,a, Yu et al., 2025, Hu et al., 2025, Zhang et al., 2025]. To foster more general and comprehensive reasoning abilities, it is essential to utilize diverse tasks and examples with verifiable rewards. In this work, we concentrate on logical reasoning as a promising domain for this objective, hypothesizing that logical reasoning serves as a fundamental building block for developing general reasoning skills. Although prior work

39th Conference on Neural Information Processing Systems (NeurIPS 2025).

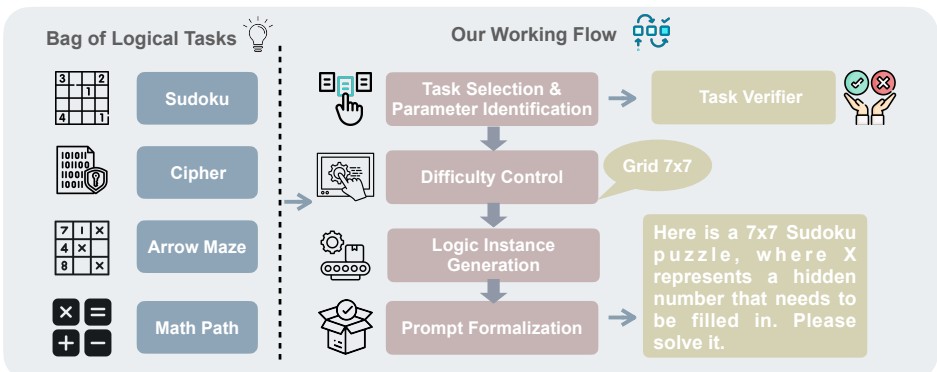

Figure 1: The framework of logic data synthesis. The process begins with the selection of suitable tasks and the identification of key parameters that control task difficulty. Next, logic instances are generated with appropriate difficulty control (e.g., setting the grid size of Sudoku to 7). These instances are subsequently formalized into natural language instructions. Each task is paired with a task-specific verifier to check the correctness of responses. This framework enables the systematic synthesis of high-quality logic data, covering a wide range of difficulty levels and 35 task types.

has explored RL training in the context of logic tasks [Xie et al., 2025b, Pan et al., 2025, Tong et al., 2025, Jiang et al., 2025], these efforts have typically focused on a single task, leaving the potential of broader and more diverse synthetic logic datasets largely underexplored.

Synthetic logic data presents distinct advantages and challenges. Its synthetic nature allows for unlimited data generation with controllable difficulty levels, enabling the creation of increasingly challenging samples. Additionally, the intrinsic properties of some logic tasks like Sudoku often require trial and backtracking in the reasoning process, which closely relates to the "aha moments" [DeepSeek-AI et al., 2025] in problem-solving. Therefore, the primary advantages of synthetic logic data for RLVR lie in its scalability and inherent characteristics that align well with complex reasoning processes. The main challenge, however, is the complexity of generating and designing specific rules for different logic tasks, as tasks like Game of 24 and Sudoku each require distinct verifiers.

Recent works primarily focus on logic evaluation [Suzgun et al., 2022, Ma et al., 2024, Kazemi et al., 2025], but lack high-quality accessible logical reasoning training data. In this work, to address the gap in comprehensive logic tasks, we present SYNLOGIC: a logical reasoning data synthesis framework and a comprehensive synthetic logic dataset containing 35 tasks, including typical logical tasks such as Sudoku, Game of 24, and Cipher. For each task, we develop task-specific generation code paired with a corresponding rule-based verifier, allowing for fine-grained difficulty control through adjustable generation hyperparameters.

To validate the effectiveness of reinforcement learning on the SYNLOGIC data, we run RL training on it with the GRPO algorithm [Shao et al., 2024] and implement binary outcome rewards determined by each task's verification rules. By adapting recent GRPO training techniques introduced in DAPO [Yu et al., 2025], we successfully train Qwen2.5 Base models [Yang et al., 2024] on the SYNLOGIC data in a zero RL training setting, achieving progressively longer COT responses and observing the emergence of reflection behaviors. Starting from Qwen2.5-7B-Base and Qwen2.5-32B-Base foundations, our models achieve over 8 absolute percentage points improvement on the logic benchmark KOR-Bench [Ma et al., 2024] compared to their instruction models. Notably, our 32B model surpasses DeepSeek-R1-Distill-Qwen-32B on BBEH [Kazemi et al., 2025] tasks by 5 absolute points, establishing SYNLOGIC as the state-of-the-art open-source dataset for logical reasoning to date. Additionally, both models demonstrate strong generalization to unseen mathematics domains. Beyond the Qwen family, we also conduct experiments on OctoThinker-8B-Base [Wang et al., 2025], a LLaMA-v3.1-8B [AI@Meta, 2024] mid-trained model, observing similar success.

Furthermore, we explore mixing the SYNLOGIC data with mathematics or coding data for RL training. Surprisingly, conducting the mixed training on Qwen2.5-7B-Base model [Yang et al., 2024], incorporating SYNLOGIC data improves training efficiency for developing mathematical and coding skills. For mathematics, mixed training maintains similar mathematics performance under the same

Table 1: Comparison of SYNLOGIC with existing synthetic logic datasets. *The number of tasks of KOR-Bench is based on the broader categorization in the paper. "Trainable" indicates whether the dataset provides training data.

| Dataset | Tasks | Trainable | Adjustable Difficulty |
|---|---|---|---|
| BBH [Suzgun et al., 2022] | 23 | ✗ | ✗ |
| Zebra Logic [Lin et al., 2024] | 1 | ✗ | ✓ |
| KOR-Bench [Ma et al., 2024] | 5* | ✗ | ✗ |
| K&K [Xie et al., 2025a] | 1 | ✓ | ✓ |
| BBEH [Kazemi et al., 2025] | 23 | ✗ | ✗ |
| SYNLOGIC | 35 | ✓ | ✓ |

number of training steps, which consume fewer math training samples. Simultaneously, mixed training achieves much higher performance on logic tasks. A similar trend is observed when mixing SYNLOGIC with coding data, further demonstrating the complementary benefits of logical reasoning training. Finally, we conduct large-scale mixed training on the Qwen2.5-32B-Base model to enhance the capability of Zero-RL training. Our mixed training achieves superior performance on multiple benchmarks compared to the DeepSeek-R1-Zero-Qwen-32B model, consistently outperforming or matching it on BBEH [Kazemi et al., 2025], KOR-Bench [Ma et al., 2024], LiveCodeBench [Jain et al., 2025], and GPQA-Diamond [Rein et al., 2024], validating the generalization benefits provided by the inclusion of logical reasoning data.

## 2 SYNLOGIC: Synthesizing Logical Reasoning Data at Scale

### 2.1 Background

Logical reasoning has long been a crucial indicator of model intelligence [ai2, 2019, Suzgun et al., 2022], valued for both its significance and synthetic accessibility. With the advancement of reasoning capabilities in Large Language Models (LLMs), researchers have developed increasingly challenging benchmarks to evaluate logical reasoning abilities [Kazemi et al., 2025, Ma et al., 2024]. However, as illustrated in Table 1, existing benchmarks either lack training support or are limited to a small number of tasks. Synthetic logic data serves as an important source of verifiable data and offers straightforward control over task difficulty, presenting the potential for developing scalable stronger models by training on it. Consequently, comprehensive synthetic logic datasets are essential for developing general strong reasoning models [Seed et al., 2025].

### 2.2 The Data Synthesis Framework

To synthesize large-scale, diverse synthetic data, we develop a comprehensive data synthesis framework encompassing 35 tasks. While the benchmarks in Table 1 include a wide variety of tasks, a significant challenge we faced is that nearly all evaluation benchmarks do not open-source their data generation methods. This hinders us from building training data for these logic tasks directly. Therefore, we develop these tasks independently, building the SYNLOGIC framework, illustrated in Figure 1. The framework consists of the following key components:

1. **Task Selection** We select a diverse set of logic tasks that require non-trivial reasoning, drawing from two carefully curated categories of data sources: (1) widely recognized puzzle problems from logic communities, such as the game of 24, Sudoku, and cryptarithms. Many of these puzzles have been previously highlighted in works like [Kurtic et al., 2024, Li et al., 2024, Ma et al., 2024]. (2) Logic tasks featured in established evaluation benchmarks, including BBH [Suzgun et al., 2022] and BBEH [Kazemi et al., 2025]. Detailed descriptions and sources for all 35 tasks can be found in the Appendix A.1.

2. **Parameter Identification** For each task, we identify key parameters that control difficulty, such as grid size in Sudoku or the missing number ratio in Math Path. These parameters form the basis for scalable and adjustable difficulty in data synthesis. For instance, a higher grid size in Sudoku typically represents greater difficulty, while a higher ratio of missing numbers in Math Path increases computational complexity.

3. **Logic Instance Generation** We formalize the task-specific rules into code by manually implementing rule-based logic generators for each task. These generators are designed to encode the specific constraints and rules of the logic problems, ensuring that the generated instances adhere to the intended task structure (e.g., enforcing the unique digits rule in Sudoku). This rule-based approach allows us to efficiently produce large quantities of data and to cover a broad spectrum of difficulty levels by adjusting the difficulty related parameters. All generated instances undergo automated checks for correctness and solvability.

4. **Appropriate Difficulty Control** To ensure that the generated data is both challenging and learnable, we carefully adjust difficulty-related parameters during data generation. We use strong reasoning models, DeepSeek R1 [DeepSeek-AI et al., 2025] and OpenAI-o3-mini to set an upper bound on difficulty: the highest difficulty parameters for which R1 or o3-mini can solve samples with a pass@10 greater than zero, representing the limit of these models' solvability. This approach prevents the inclusion of instances that are too difficult. Similarly, we use Qwen2.5-32B-Instruct [Yang et al., 2024] to determine the lower bound of difficulty: the lowest difficulty parameters for which the models achieve a pass rate between 0 and 0.5. This dual-bound approach ensures that the dataset includes a balanced range of samples, maintaining an appropriate level of complexity and learnability.

5. **Prompt Formalization** To facilitate training and evaluation with LLMs, we convert abstract logic instances into natural language prompts using task-specific prompt templates. To ensure robustness, we create an average of over 10 prompts per task, covering both English and Chinese versions. This approach guarantees that each instance is accessible to both humans and language models while maintaining consistency across languages and phrasings.

6. **Verification Suite** For every task, we implement a dedicated verifier that automatically checks the correctness of model outputs, supporting both RL training supervision and automatic evaluation. Our verifier validates whether the final answer is correct, rather than evaluating intermediate reasoning steps, consistent with other RLVR approaches. For example, in Sudoku, we identify the difficulty parameter as grid size (set to 7×7). Our generator creates instances with missing numbers marked as X, which are then formatted into natural language prompts. The verifier checks whether the model's completed grid complies with Sudoku rules (such as single value rules) to verify correctness. This verification approach ensures accurate and reliable assessment without apparent failing cases across all tasks in SYNLOGIC.

A key innovation in our approach is the development of customized difficulty control mechanisms for each task type. Unlike existing benchmarks that often provide fixed-difficulty evaluation data, our system allows precise calibration of problem complexity through task-specific parameters, such as grid size in Sudoku. This difficulty-tuning capability enables the creation of different difficulty level data, presenting the potential of progressively challenging training curricula. We overcame significant challenges in implementing these controls, as many evaluation benchmarks do not open-source their data generation methods. At last, most tasks in SYNLOGIC are designed with: (1) a data generation code capable of producing varied instances, (2) a corresponding verification rule for evaluating solution correctness, and (3) configurable difficulty parameters to enable controlled difficulty of generated data. We independently develop and generate data for 33 tasks in our dataset, while only the data of 2 tasks (Zebra Puzzle [Lin et al., 2024] and ARC-AGI [Chollet, 2019]) are directly adopted from existing open source resources.

**Risk of Data Contamination** Although several tasks overlap between our selected tasks and current benchmarks, such as KOR-Bench and BBEH, the synthetic nature of our data, combined with the large synthesis space, makes the probability of generating data identical to benchmark test samples very low – we have verified that there are no identical samples between our generated datasets and the benchmark test sets.

## 2.3 The SYNLOGIC Datasets

We synthesized our dataset with controlled difficulty parameters for each task, carefully balancing challenge and learnability to ensure the success of our subsequent experiments §3. To accommodate different model capacities, we developed two distinct versions of our dataset: **SYNLOGIC-Hard** for Qwen2.5-32B training and **SYNLOGIC-Easy** for Qwen2.5-7B and OctoThinker-8B [Wang et al., 2025] training. SYNLOGIC-Hard presents more complex challenges with its broader difficulty level

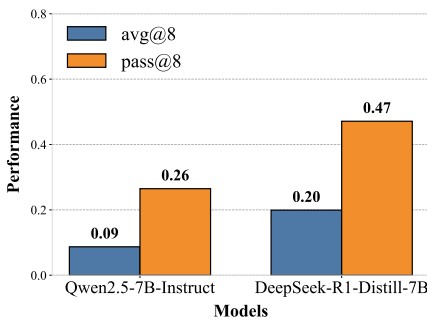
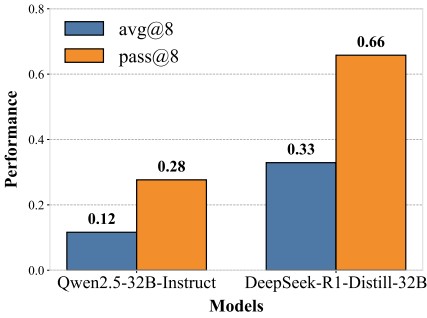

(a) 7B models on SYNLOGIC-Easy  (b) 32B models on SYNLOGIC-Hard

Figure 2: Evaluation of task difficulty across our dataset versions. (a) Shows the performance of 7B-scale models on the SYNLOGIC-Easy dataset, while (b) demonstrates the performance of 32B-scale models on the more challenging SYNLOGIC-Hard dataset. Results are measured using avg@8 (average pass rate with eight attempts) and pass@8 (success within eight attempts) metrics, illustrating the appropriate difficulty control for each model scale.

for each tasks with difficulty upper bound described in § 2.2. For SYNLOGIC-Easy, we systematically lower difficulty parameters across all tasks to create easier versions. Despite these adjustments, eight tasks still remain beyond the learning capacity of the 7B model with zero training accuracy after RL, thus we removed them from this easy version, where the details about the removed tasks are provided in Appendix A.2. Finally, we synthesized 33k SYNLOGIC-Hard samples and 16k SYNLOGIC-Easy samples used in subsequent experiments for training, along with 10 validation samples per task, separately for the Easy and Hard validation splits.

**Difficulty Analysis** To assess the difficulty of the synthetic data, we conduct an evaluation on the validation splits, assessing model performance using both avg@8 (average pass rate with eight attempts) and pass@8 (success within eight attempts) metrics. The results, illustrated in Figure 2, confirm the appropriate difficulty levels for each model scale, demonstrating that our datasets provide suitable training challenges across different model capacities.

# 3 Reinforcement Learning on SYNLOGIC

Reinforcement learning with verifiable rewards (RLVR) has emerged as a highly effective approach for enhancing reasoning capabilities in large language models [Zeng et al., 2025b, Yu et al., 2025, Hu et al., 2025, Zhang et al., 2025]. Building on these advances, our experimental framework also focuses on applying reinforcement learning techniques to the SYNLOGIC dataset, leveraging the verifiable nature of logical reasoning tasks. In this section, we validate the effectiveness of reinforcement learning training on the SYNLOGIC dataset using Qwen2.5-7B-Base, Qwen2.5-32B-Base, and OctoThinker-8B-Base, a LLaMA-v3.1-8B mid-trained model.

## 3.1 Setup Details

**Training Template** Following the DAPO training prompt [Yu et al., 2025], we modify and design the training prompt template for logic training as shown in Figure 3:

**Reward Design** Our reward function employs a binary scoring mechanism that evaluates both format adherence and answer correctness. Specifically, we assign a reward of 1 only when a model-generated response satisfies two criteria: (1) it correctly follows the designated format by including both `<think>` `</think>` and `<answer>` `</answer>` tags, and (2) the final answer provided is correct. Responses that either deviate from the required format or contain incorrect answers receive a reward of 0.

> **SYNLOGIC Training Prompt Template**
>
> Solve the following problem step by step. First, think about the reasoning process in the mind and then provide the answer. The reasoning process is enclosed within <think> </think> and the final answer is enclosed within <answer> </answer> tags, respectively, i.e., <think> reasoning process here </think> <answer> answer here</answer>.

Figure 3: The prompt template used for training models on SYNLOGIC data.

Table 2: Evaluation accuracy (%) of 7B, 32B and 8B models across logic benchmarks (SYNLOGIC-Val, KOR-Bench, BBH, BBEH) and mathematical benchmarks (AIME 2024, MATH 500, AMC 2023). For 7B and 8B models, SYNLOGIC-Val refers to SYNLOGIC-Easy, while for 32B models, it refers to SYNLOGIC-Hard as described in § 2.3. All evaluations were conducted under zero-shot conditions, with SYNLOGIC-Val and AIME 2024 reported as avg@8 to reduce variance.

| Model | Logic Benchmarks | | | | Mathematical Benchmarks | | |
|---|---|---|---|---|---|---|---|
| | SYNLOGIC-Val | KOR-Bench | BBH | BBEH | AIME 2024 | MATH 500 | AMC 2023 |
| Qwen2.5-7B-Base | 2.8 | 11.6 | 45.2 | 3.8 | 0.3 | 64.6 | 30.0 |
| Qwen2.5-7B-Instruct | 9.0 | 38.6 | 62.7 | **12.4** | 6.3 | **76.4** | 52.5 |
| SYNLOGIC-7B | **44.4** | **48.1** | **66.5** | 8.0 | **10.0** | 71.8 | **55.0** |
| Qwen2.5-32B-Base | 1.6 | 10.9 | 58.4 | 3.3 | 4.5 | 68.6 | 45.0 |
| Qwen2.5-32B-Instruct | 12.0 | 54.7 | 84.5 | 17.5 | 10.0 | 82.2 | 57.5 |
| R1-Distill-Qwen-32B | 33.0 | **66.6** | **88.3** | 19.2 | **72.6** | **94.3** | **85.0** |
| SYNLOGIC-32B | **52.9** | 62.2 | 85.8 | **25.5** | 19.6 | 82.0 | 57.5 |
| OctoThinker-8B (Base) | 4.2 | 37.2 | 43.5 | 4.1 | 3.3 | 38.7 | 20.3 |
| SYNLOGIC-8B | **38.0** | **44.0** | **55.2** | **12.0** | **14.5** | **68.4** | **45.0** |

$$R(o_i) = \begin{cases} 1, & \text{if format}(o_i) = \text{True} \ \wedge \ \text{correct}(o_i) = \text{True} \\ 0, & \text{otherwise} \end{cases} \tag{1}$$

where $\text{format}(o_i)$ evaluates whether response $o_i$ includes both the required <think> </think> and <answer> </answer> tags, and $\text{correct}(o_i)$ determines whether the answer provided is correct verified by its task's verification rule.

**Training Details**  For our experiments, we synthesized approximately 16k SYNLOGIC-Easy and 33k SYNLOGIC-Hard instances to train the Qwen2.5-7B-Base, OctoThinker-8B-Base, and Qwen2.5-32B-Base models with DAPO, respectively, as described in §2.3. During training, we employed a prompt batch size of 128, generated 16 rollouts per prompt, and set maximum rollout lengths of 16,384 tokens for the 7B model and 28,672 tokens for the 32B model. We configured the clip high parameter $\epsilon_{\text{high}}$ at 0.28. Additional training hyperparameters and implementation details are provided in Appendix B.1.2.

**Evaluation Details**  Our evaluation strategy encompasses two distinct benchmark categories. For assessing logical reasoning capabilities, we employ the validation splits of SYNLOGIC alongside established benchmarks including Knowledge-Orthogonal Reasoning (KOR-Bench) [Ma et al., 2024], BBH [Suzgun et al., 2022], and the substantially more challenging BBEH [Kazemi et al., 2025]. To investigate cross-domain generalization effects, we incorporate mathematics evaluations on MATH 500 [Hendrycks et al., 2021], AMC 2023, and AIME 2024. All evaluations are conducted in a zero-shot setting, with avg@8 metrics computed for AIME 2024 and SYNLOGIC-Val to mitigate variance.

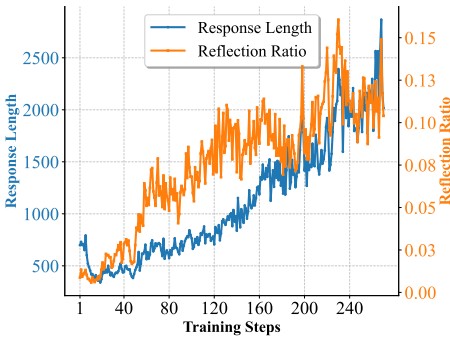 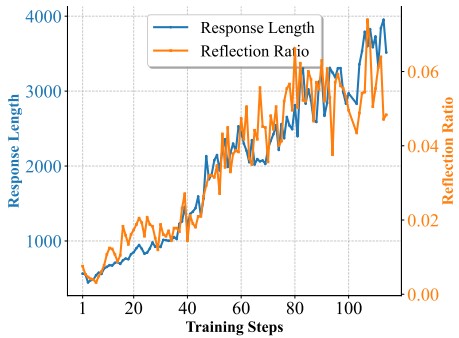

(a) Avg Length and Reflection of 7B Training.  (b) Avg Length and Reflection of 32B Training.

Figure 4: Response length and reflection ratio across the 7B and 32B training process on the training dataset. The reflection ratio represents the proportion of generated responses containing at least one reflection phrase (including "recheck", "rethink", "try again", "let's correct it", "re-evaluate", "check again", "think again").

## 3.2 Results

**Substantial Improvements in Logical Reasoning**  The evaluation results presented in Table 2 demonstrate significant improvements across logical reasoning tasks. Beyond the notable gains on SYNLOGIC's validation split, our models demonstrate enhanced performance across multiple logical benchmarks, leading state-of-the-art results among open-source datasets. Our 7B model achieves 48.1% on KOR-Bench [Ma et al., 2024], outperforming Qwen2.5-7B-Instruct by nearly 10 absolute percentage points. Similarly, our 32B model surpasses Qwen2.5-32B-Instruct by 7 percentage points on KOR-Bench. Notably, our 32B model exceeds R1-Distill-Qwen32B [DeepSeek-AI et al., 2025] by 6 percentage points on the challenging BBEH benchmark [Kazemi et al., 2025], showcasing the effectiveness of the SYNLOGIC dataset in driving state-of-the-art logical reasoning performance. Beyond the Qwen family, our substantial improvements on the 8B model also validate the effectiveness of our method on LLaMA base models.

**Generalization to Mathematical Domains**  Our experimental results demonstrate significant generalization capabilities to mathematical domains, as shown in Table 2. Despite being primarily trained for logical reasoning, SYNLOGIC models exhibit strong performance across mathematical benchmarks over their base models. SYNLOGIC-7B achieves 10.0% on AIME 2024, a nearly 10 absolute point improvement compared to Qwen2.5-7B-Base (0.3%), 71.8% on MATH 500, a 7.2 absolute point gain, and 55.0% on AMC 2023, a 25-point increase. More remarkably, SYNLOGIC-32B achieves 19.6% on AIME 2024, a 4.4x improvement over Qwen2.5-32B-Base (4.5%), while its performance on MATH 500 (82.0%) and AMC 2023 (57.5%) shows substantial gains of 13.4 and 12.5 points, respectively. Similarly, SYNLOGIC-8B exhibits strong mathematical capabilities, achieving 14.5% on AIME 2024, with 11.2 absolute improvement. Without mathematics training data, our models nearly match or surpass the instruction models, suggesting that enhancements in logical reasoning capabilities transfer effectively to mathematical problem-solving. This aligns with the observation in Logic-RL [Xie et al., 2025b], highlighting the fundamental connection between logical and mathematical reasoning skills.

**Increased Chain-of-Thought Length**  As shown in Figure 4, recording the response length and reflection ratio during the training process reveals that training on SYNLOGIC data leads to stable increases in response length for both models. The 7B model reaches an average of approximately 2500 tokens, while the 32B model achieves around 4000 tokens. Additionally, the increasing reflection ratios also indicate the emergence of cognitive behaviors during training [Gandhi et al., 2025]. Both the extended response lengths and increased prevalence of reflection tokens suggest that synthetic logic reasoning tasks inherently align with the long-thinking paradigm of LLM. Detailed training accuracy results are provided in Appendix B.1.3.

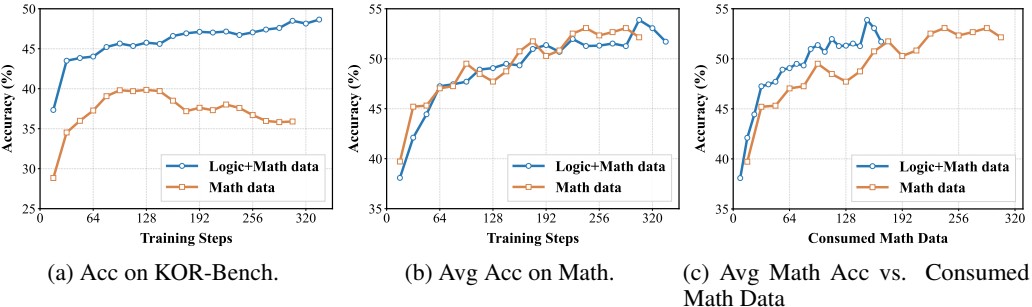

| (a) Acc on KOR-Bench. | (b) Avg Acc on Math. | (c) Avg Math Acc vs. Consumed Math Data |

Figure 5: Performance comparison of 7B models trained on mixed data (Logic+Math) versus math-only (Math) data. (a) Accuracy on KOR-Bench. (b) Average accuracy across three mathematics benchmarks (MATH 500, AIME 2024, AMC 2023) as a function of training steps. (c) Average accuracy on mathematics benchmarks as a function of consumed mathematical data volume.

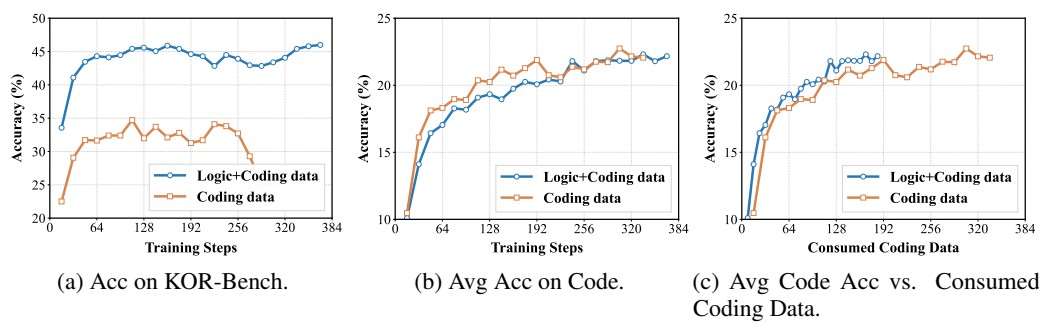

| (a) Acc on KOR-Bench. | (b) Avg Acc on Code. | (c) Avg Code Acc vs. Consumed Coding Data. |

Figure 6: Performance comparison of 7B models trained on mixed data (Logic+Coding) versus coding-only (Coding) data. (a) Accuracy on KOR-Bench. (b) Average accuracy across two coding benchmarks (validation split of our coding data and LiveCodeBench [Jain et al., 2025]) as a function of training steps. (c) Average accuracy on coding benchmarks as a function of consumed coding data volume.

## 4 Scaling RL Training with Diverse Verifiable Reasoning Data

Having verified the success of reinforcement training on SYNLOGIC alone, we now leverage verifiable reasoning data from math, coding, and logical reasoning domains, scaling RL training with diverse verifiable reasoning data. Concretely, we mix SYNLOGIC with mathematical or/and coding training data, and perform RL training on the mixed datasets. We will study how combining logical reasoning with code and mathematical data influences training efficiency on 7B models and enhances the Zero-RL capabilities for 32B models.

### 4.1 Setup Details

For mathematical training data, we directly utilize the 17k samples provided in DAPO [Yu et al., 2025]. For coding data, we assembled approximately 9k samples from various online coding platforms such as Codeforces. We adapted a similar prompt template to that used in our SYNLOGIC training; the detailed templates are presented in Appendix B.1. We maintained the same reward design approach as described in §3.1. Specifically, for coding tasks, the reward is set as 1 if the output is correctly formatted and all test cases pass; otherwise, the reward is 0.

### 4.2 Mixing SYNLOGIC with Math or Code Data: A Pilot Ablation Study

**Mixed Training with Math**   We sample approximately 17k samples from SYNLOGIC-Easy and combine them with 17k math data for training on the Qwen2.5-7B-Base model. For controlled comparison, we also conduct reinforcement learning using exclusively math data. Both experimental

Table 3: Performance comparison across multiple benchmarks. The evaluation metrics vary by dataset: BBEH [Kazemi et al., 2025] uses pass@1, while KOR-Bench [Ma et al., 2024], LiveCodeBench (LCB)[Jain et al., 2025], and GPQA-Diamond[Rein et al., 2024] use avg@4. AIME 2024 is evaluated using avg@8. All training configurations (Zero-Mix-2 and Zero-Mix-3) are run for the same number of training steps to ensure a fair comparison of results.

| Model | BBEH | KOR-Bench | LCB | AIME 2024 | GPQA Diamond |
|---|---|---|---|---|---|
| DeepSeek-R1-Distill-Qwen-32B | 19.2 | 66.6 | 57.2 | 72.6 | 63.1 |
| DeepSeek-R1-Zero-Qwen-32B | - | - | 40.2 | **47.0** | 55.0 |
| Zero-Mix-2 (Math+Coding) | 18.5 | 58.6 | 39.5 | 34.5 | 55.2 |
| Zero-Mix-3 (SYNLOGIC+Math+Coding) | **28.6** | **65.0** | **40.7** | 35.8 | **57.5** |

configurations maintain identical hyperparameters, optimization settings, and computational resources to ensure fair evaluation. Figure 5 presents a comparison of training dynamics. Running for the same number of training steps, mixed training (Logic+Math) achieves comparable performance to math-only training on average across three mathematical benchmarks (Figure 5b), while consuming fewer math samples. Under the same volume of processed math data, mixed training achieves higher accuracy (Figure 5c). Moreover, mixed training steadily improves logical reasoning, as reflected in rising KOR-Bench scores (Figure 5a), which are nearly 10 absolute percentage points higher than those achieved with math-only training. These results suggest that mixed training facilitates more efficient optimization, potentially due to shared abstract reasoning mechanisms across domains.

**Mixed Training with Code**  Following a similar methodology, we sample approximately 9k samples from SYNLOGIC-Easy and combine them with 9K code samples to train the Qwen2.5-7B-Base model. As a control, we conduct parallel training using exclusively coding data. Both training configurations maintain identical parameters to ensure a fair comparison. To measure the coding ability, we include the validation split of our coding data and the LiveCodeBench [Jain et al., 2025] (the same version as used in DeepSeek's report [DeepSeek-AI et al., 2025]) for evaluation. As shown in Figure 6, we observe a similar phenomenon of more efficient training dynamics when mixing code with SYNLOGIC-Easy. Models trained on Logic+Coding data achieve higher performance on coding benchmarks than code-only training when consuming the same volume of coding data. Simultaneously, mixed training improves logical reasoning, as evidenced by 10 absolute points better KOR-Bench scores (Figure 6a). These findings reinforce the complementary nature of logical reasoning in enhancing domain-specific capabilities.

### 4.3  32B Zero-RL Training with Diverse Reasoning Data

Building on the previous observation, here we scale up the diverse, verifiable training data by mixing math, coding, and SYNLOGIC datasets, and perform RL training on the Qwen2.5-32B-Base model. Specifically, we use a mix of 35k mathematical samples, 9k coding samples, and 17k SYNLOGIC samples for training. We term this training configuration as **Zero-Mix-3**. We additionally conduct a **Zero-Mix-2** setting that only mixes coding and mathematical data, serving as an ablation baseline to study the effect of SYNLOGIC in such a scalable setting. Related to our **Zero-Mix-2** setup, Zhang et al. [2025] recently demonstrated that combining mathematical data with coding data is able to facilitate coding learning. Here we further scale up this trend by including our proposed SYNLOGIC dataset. To evaluate generalization, we include an out-of-domain benchmark, GPQA Diamond [Rein et al., 2024], to study how the addition of SYNLOGIC impacts broader reasoning capabilities. Both Zero-Mix-3 and Zero-Mix-2 configurations are run for the same number of training steps to ensure a fair and controlled comparison.

**Results**  As shown in Table 3, the Zero-RL training in **Zero-Mix-3** (SYNLOGIC+Math+Coding) achieves superior performance across multiple evaluations. On logic benchmarks, Zero-Mix-3 nearly matches the performance of DeepSeek-R1-Distill-Qwen-32B on KOR-Bench and surpasses it by 8 points on BBEH. Notably, Zero-Mix-3 also matches DeepSeek-R1-Zero-Qwen-32B on the coding benchmark LiveCodeBench [Jain et al., 2025] and outperforms it on GPQA-Diamond. Compared to the **Zero-Mix-2** (Math+Coding) experiment, Zero-Mix-3 consistently delivers higher performance

across all benchmarks. Specifically, Zero-Mix-3 shows a significant improvement of over 10 points on BBEH, 6 points on KOR-Bench, and over 2 points on the out-of-domain benchmark GPQA Diamond. These results strongly validate the significant generalization benefits provided by the inclusion of SYNLOGIC.

## 5    Conclusion

We present SYNLOGIC: a data synthesis framework and a comprehensive synthetic logic dataset with 35 diverse tasks, addressing the lack of high-quality logic training data. Using SYNLOGIC, we trained Qwen2.5 models with the GRPO algorithm, achieving significant gains on logic benchmarks like KOR-Bench and strong generalization to unseen mathematical tasks. Notably, our 32B model outperformed DeepSeek-R1-Distill-Qwen-32B on BBEH. Mixed training with SYNLOGIC further improved training efficiency and performance, showcasing the complementary benefits of logical reasoning across domains. We hope SYNLOGIC inspires broader exploration of synthetic datasets and logical reasoning to develop stronger reasoning capability models.

**Limitations**    First, due to limited computational resources, we do not conduct precise difficulty tuning for every task, meaning the curated data may not be fully optimized for training 7B and 32B models. Second, we do not implement training with dynamically adjustable difficulty, where sample complexity progressively increases during training. We hypothesize that such an approach could further enhance the logical reasoning capabilities of LLMs. Both of these directions are left as opportunities for future exploration.

## Acknowledgements

This project is partially supported by Hong Kong RGC ECS grant no. 26218125 and NSFC grant no. 62306177.

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

# A Comprehensive Overview of SYNLOGIC

## A.1 Task Composition and Sources

Table 4 presents the diverse collection of tasks incorporated in SYNLOGIC. We have carefully selected these tasks from established benchmarks including KOR-Bench [Ma et al., 2024], BBH [Suzgun et al., 2022], and BBEH [Kazemi et al., 2025]. Additionally, we integrated some logical reasoning tasks not previously featured in these benchmarks, such as Mathador [Kurtic et al., 2024] and Minesweeper [Li et al., 2024].

Our collection comprises 35 distinct tasks, with only two (Zebra Puzzle [Lin et al., 2024] and ARC-AGI [Chollet, 2019]) using existing data sources. For all remaining tasks, we generated custom datasets. Importantly, we developed and implemented verifiers for all tasks in the collection, ensuring consistent evaluation across the benchmark.

Table 4: Tasks in the Dataset Collection with Descriptions

| No. | Task Name | Description |
| --- | --- | --- |
| 1 | ARC-AGI | A collection of general intelligence tasks requiring abstract reasoning and pattern recognition. |
| 2 | Arrow Maze | A maze-solving task where arrows dictate movement, requiring pathfinding logic. |
| 3 | Boolean Expressions | Evaluating logical expressions with AND, OR, NOT operators. |
| 4 | Buggy Tables | Correcting flawed data in tables based on logical constraints. |
| 5 | Calcudoko | A math-based Sudoku variant with arithmetic constraints. |
| 6 | Campsite | Placing tents in a grid while satisfying adjacency rules. |
| 7 | Cipher | Decoding encrypted messages based on given rules. |
| 8 | Cryptarithm | Solving arithmetic puzzles where letters represent digits. |
| 9 | Dyck Language | Validating bracket sequences for correct nesting. |
| 10 | Dyck Language Errors | Identifying and correcting errors in bracket sequences. |
| 11 | Dyck Language Reasoning Errors | Advanced reasoning for errors in bracket nesting logic. |
| 12 | Futoshiki | A grid-based logic puzzle with inequality constraints. |
| 13 | Goods Exchange | Tracking item exchanges among multiple participants. |
| 14 | Kukurasu | A grid-based puzzle involving row/column weight sums. |
| 15 | Mathador | A math strategy game involving arithmetic operations. |
| 16 | Math Path | Finding correct numbers to satisfy equations in a grid. |
| 17 | Minesweeper | Logical deduction to uncover mines on a grid. |
| 18 | Norinori | Placing domino tiles in a grid with adjacency rules. |
| 19 | Number Wall | Constructing walls to separate grid regions based on rules. |
| 20 | Numbrix | Filling a grid with consecutive numbers in order. |
| 21 | Object Counting | Counting specific objects under given constraints. |
| 22 | Object Properties | Inferring and reasoning about object attributes. |
| 23 | Operation | Solving puzzles with custom-defined mathematical operations. |
| 24 | Skyscraper Puzzle | Determining building heights based on visibility clues. |
| 25 | Space Reasoning | Reasoning about spatial relationships in a grid. |
| 26 | Space Reasoning Tree | Advanced spatial reasoning tasks with hierarchical relationships. |
| 27 | Star Placement Puzzle | Placing stars in a grid while avoiding adjacency conflicts. |
| 28 | Sudoku | Solving the classic number-placement puzzle. |
| 29 | Survo | Filling grids to satisfy row and column sum constraints. |
| 30 | Time Sequence | Scheduling tasks or events with overlapping constraints. |
| 31 | Web of Lies | Determining truth-tellers and liars through logical statements. |
| 32 | Word Sorting | Sorting words based on custom rules or constraints. |
| 33 | Word Sorting Mistake | Identifying mistakes in word sorting logic or reasoning. |
| 34 | Wordscapes | A crossword puzzle where players fill words from lists while matching intersections. |
| 35 | Zebra Puzzle | Solving complex logic puzzles with multiple constraints. |

## A.2 SYNLOGIC-Hard and SYNLOGIC-Easy

The SYNLOGIC-Hard dataset encompasses all 35 tasks, representing a challenging upper bound calibrated to the solvability thresholds of DeepSeek R1 and OpenAI-o3-mini. During our experiments with Qwen2.5-32B-Base, we observed consistent training accuracy gains across this comprehensive task set. However, this difficulty level proved excessive for smaller-scale models like Qwen2.5-7B-Base. Despite reducing the difficulty parameters across tasks, eight specific tasks (Arrow Maze, Goods Exchange, Kukurasu, Minesweeper, Norinori, Object Counting, Space Reasoning Tree, and Wordscapes) persistently yielded zero accuracy when training the 7B models. Consequently, we developed SYNLOGIC-Easy, a modified variant that excludes these particularly challenging tasks, to provide a more appropriate training dataset for the Qwen2.5-7B-Base model.

# B   Training and Evaluation Details

## B.1   Training

### B.1.1   Training Template

We provide the training template for math and coding here in Figure 7 and Figure 8.

> **Math Training Prompt Template**
>
> You are a helpful assistant. You always first think about the reasoning process in the mind and then provides the user with the answer.\nThe reasoning process and answer are enclosed within '<think>' '</think>' and '<answer>' '</answer>' tags, respectively, e.g.,\n<think>\nA detailed reasoning process here, with possible reflections including but not limited to reviewing previous steps for errors, exploring alternative approaches, and considering possible refinements.\n</think>\n<answer>\nReply to user here.\n</answer>. Please reason step by step, and put your final answer within \boxed{}.

Figure 7: The prompt template used for training models on math data.

> **Coding Training Prompt Template**
>
> You are a helpful assistant. You always first think about the reasoning process in the mind and then provides the user with the answer.\nThe reasoning process and answer are enclosed within '<think>' '</think>' and '<answer>' '</answer>' tags, respectively, e.g.,\n<think>\nA detailed reasoning process here, with possible reflections including but not limited to reviewing previous steps for errors, exploring alternative approaches, and considering possible refinements.\n</think>\n<answer>\nReply to user here.\n</answer>. Please put your final code following this format:\n```[language]\n#your code here \n```.

Figure 8: The prompt template used for training models on coding data.

### B.1.2   Training hyper-parameters

For 7B model training, we adopted the following hyper-parameters: with learning rate 1e-6, GRPO group size 16, max prompt length 2048, max response length 16384, prompt batch size 128, mini batch size 64, with clip high 0.28, clip low 0.2.

For 32B model training in §3, we adopted the following hyper-parameters: with learning rate 2e-6, GRPO group size 16, max prompt length 2048, max response length 28672, prompt batch size 128, mini batch size 16, with clip high 0.28, clip low 0.2.

For 32B model training in §4, we adopted the following hyper-parameters: with learning rate 2e-6, GRPO group size 16, max prompt length 2048, max response length 12288, prompt batch size 512, mini batch size 16, with clip high 0.28, clip low 0.2.

### B.1.3   Training Dynamics Analysis

We present the training accuracy progression of our 7B and 32B models on the SYNLOGIC dataset in Figure 9. The results demonstrate a consistent improvement in accuracy throughout the training process for both model sizes.

## B.2   Evaluation

### B.2.1   Performance Analysis of Mixed Training with Math

We analyze the training dynamics when combining our logical reasoning dataset with mathematical content. Figure 10 presents performance results across three mathematical benchmarks: MATH 500, AIME 2024, and AMC 2023. These results demonstrate that mixed training can achieve similar performance with the same number of training steps while using less mathematical data. This suggests that mixed training creates more efficient training dynamics.

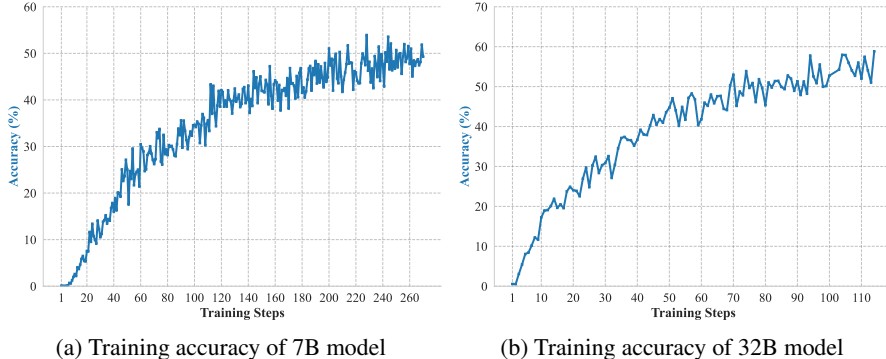

(a) Training accuracy of 7B model      (b) Training accuracy of 32B model

Figure 9: Training accuracy progression for both model sizes on the SYNLOGIC dataset. Both models exhibit steady improvement in performance as training progresses.

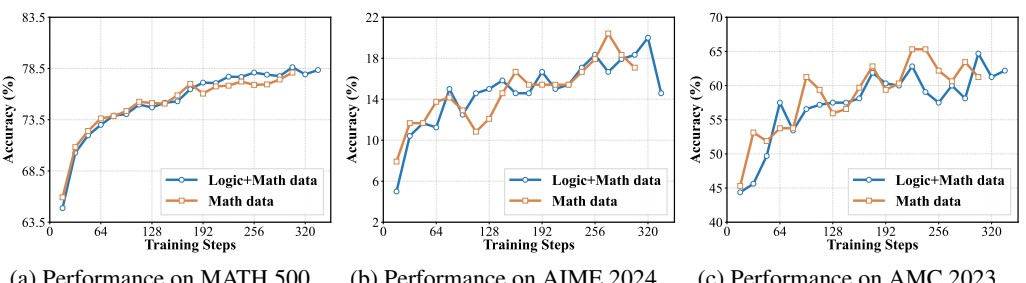

(a) Performance on MATH 500.    (b) Performance on AIME 2024.    (c) Performance on AMC 2023.

Figure 10: Comparative accuracy (%) of models trained with mixed data and math-only data across three mathematical benchmarks: MATH 500, AIME 2024, and AMC 2023. All evaluations of the figure use avg@8 scoring.

## B.3 Performance Analysis of Mixed Training with Coding

We also provide the training dynamics when mixed training with coding data in Figure 11. The observation is similar to mixed training with math, suggesting that mixed training leads to more efficient training dynamics.

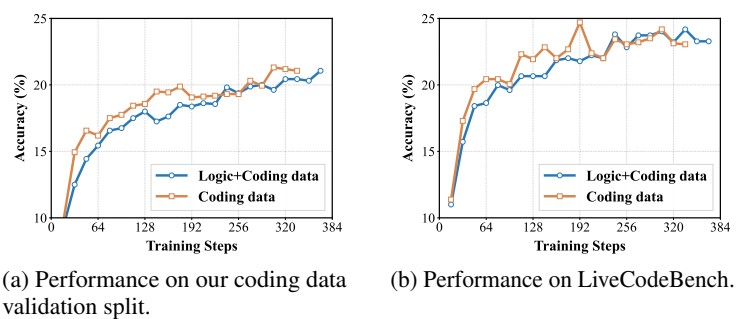

(a) Performance on our coding data    (b) Performance on LiveCodeBench.
validation split.

Figure 11: Comparative accuracy (%) of models trained with mixed data and coding-only data across two coding benchmarks: our coding data validation split and LiveCodeBench. All evaluations of the figure use avg@8 scoring.

