# OpenReview forum: "SynLogic: Synthesizing Verifiable Reasoning Data at Scale for Learning Logical Reasoning and Beyond"
_NeurIPS.cc/2025/Conference — NeurIPS 2025 poster_

### Official Review · Reviewer_cYNo · 2025-06-20

**Clarity:** 3
**Significance:** 2
**Originality:** 3
**Rating:** 4
**Confidence:** 3

**Summary:**

This paper introduces SynLogic, a framework for synthesizing large-scale, verifiable logical reasoning data across 35 diverse task types with controllable difficulty, enabling RLVR. The authors train Qwen2.5-7B and 32B models using this data, demonstrating substantial improvements on logic benchmarks like KOR-Bench and BBEH, surpassing previous SOTA open-source models. They further show that mixing SynLogic data with math and coding tasks enhances training efficiency and generalization across domains, including mathematics and programming.

**Questions:**

1. In my view, the value of high-quality synthetic data should ideally be demonstrated in the regime where strong models are already saturated with existing training data, and the synthetic data enables further gains. While Section 4.3 partially addresses this through a Zero-RL setup on Qwen2.5-32B, I would encourage the authors to include or discuss results where SynLogic data is used to fine-tune even stronger models, to evaluate whether synthetic logic tasks provide meaningful additional signal beyond what distilled data already offers.
2. While the authors demonstrate meaningful generalization improvements over base models in math and coding tasks, it would be more informative to compare against the instruct-tuned versions (e.g., Qwen2.5-Instruct), especially given the strong performance those models already exhibit in these domains. This would clarify whether SynLogic training offers significant cross-domain benefits beyond what instruction tuning alone achieves.

**Ethical Concerns:**

["NO or VERY MINOR ethics concerns only"]

**Final Justification:**

The responses have addressed my concerns.

**Limitations:**

The authors rightly acknowledge in the Limitations section that they do not implement training with dynamically adjustable difficulty. Given that SynLogic includes fine-grained difficulty control for each task, this opens up a promising direction for curriculum learning, where samples of increasing complexity could be introduced progressively. Incorporating such a strategy may further improve model performance on harder reasoning tasks, akin to how humans build up reasoning skills through gradual exposure. I encourage the authors to explore this in future work, as it aligns naturally with the design of their dataset and could enhance the effectiveness of RL training.

**Quality:**

3

**Strengths And Weaknesses:**

Strengths
1. The SynLogic dataset is designed with adjustable difficulty parameters.
2. The authors demonstrate that mixing SynLogic data with mathematical and coding datasets improves training efficiency and performance across all domains.
3. Reinforcement learning on SynLogic significantly boosts the performance of the Qwen2.5-32B model, bringing it close to the level of the state-of-the-art R1-Distill-Qwen-32B on challenging logic benchmarks such as BBEH.


Weaknesses
1. While the SynLogic dataset provides verifiable and diverse task types, the solution traces appear to be rule-based and may lack the richness and diversity of human-like reasoning strategies. Prior work [1] has shown that such diversity in solution paths plays a crucial role in improving LLM reasoning abilities. I would encourage the authors to explore or discuss approaches for enhancing the diversity of reasoning trajectories, which may further boost generalization beyond structured synthetic tasks.
2. Another concern is that the training contains only fixed-format prompts with strict reward constraints tied to specific tags. This design may inadvertently increase prompt sensitivity, a known weakness of LLMs, where slight variations in prompt phrasing can lead to large performance drops. The paper would benefit from an empirical analysis of prompt robustness or a discussion on potential mitigation strategies (e.g., training with paraphrased prompt templates or evaluating on alternate phrasing).
3. The paper highlights difficulty control as a key advantage of the SynLogic dataset, but it lacks concrete examples or definitions for how difficulty is parameterized and adjusted across all tasks. Could the authors provide detailed descriptions (possibly in Appendix) of how difficulty is defined and controlled for each task? For instance, what are the concrete parameters, value ranges, and examples across difficulty levels?
4. The verifier design is central to the reliability of RLVR. However, the paper does not elaborate on how the verifiers ensure all generated questions are solvable and whether the verifiers cover edge cases or complex failure modes. It is also unclear whether the verifiers only check for final answer correctness, or if they also assess the logical validity of intermediate reasoning steps (especially relevant under the chain-of-thought paradigm). Clarification and evidence of thorough verifier validation would significantly strengthen the work.
5. The proposed SynLogic framework demonstrates strong improvements in logical reasoning tasks when applied to Qwen2.5 models. However, all experiments are limited to the Qwen model family. Including additional results on a distinct open-source model, such as LLaMA or Mistral, would strengthen the paper’s claim beyond Qwen-specific training dynamics.

[1] Peng, Xiangyu, et al. "ReGenesis: LLMs can Grow into Reasoning Generalists via Self-Improvement." In ICLR 2025.

---

> ### Author Rebuttal · Authors · 2025-07-31
>
> Thank you for the comments and we address them below.
>
>
> >Q1: While the SynLogic dataset provides verifiable and diverse task types, the solution traces appear to be rule-based and may lack the richness and diversity of human-like reasoning strategies.
>
> A1: We agree that logical tasks are more rule-complete, but this is determined by these tasks' inherent properties. However, this does not mean these tasks don’t trigger diverse, human-like reasoning strategies to solve. As demonstrated in Figure 4, during RL training, the ratio of reflection tokens within CoT increases, which is closely related to human-like reasoning processes involving trial and backtracking. We will provide example rollouts in the appendix in the next version to further illustrate this point.
>
>
> >Q2: Another concern is that the training contains only fixed-format prompts with strict reward constraints tied to specific tags.
>
> A2: First, we do not use a single fixed prompt for each task. Instead, we implement over 10 bilingual (English and Chinese) prompts for each task separately to achieve robust prompt-following performance.  We will provide more details about the prompt examples and prompt numbers in the next version. Every generated instance is randomly selected from prompt templates to achieve diverse prompt formats and robust instruction following. Additionally, given that our approach significantly improves performance on diverse benchmarks like BBEH and KOR-Bench, it demonstrates the effectiveness of our training dataset.
>
>
> >Q3: The paper highlights difficulty control as a key advantage of the SynLogic dataset, but it lacks concrete examples or definitions for how difficulty is parameterized and adjusted across all tasks.
>
> A3: Typically, logical tasks contain key parameters that relate to difficulty or complexity. For example, the grid size of Sudoku problems directly correlates with complexity - higher grid sizes correspond to higher difficulty levels. Due to space limitations, we only showed this in Figure 1 and briefly discussed it in lines 102-104. We will definitely include more detailed descriptions of difficulty parameters and value ranges for every task in the appendix of the next version. Figure 1 provides a concrete difficulty example using Sudoku grid size.
>
>
>
> >Q4: Clarification and evidence of thorough verifier validation would significantly strengthen the work.
>
> A4: Thank you for the suggestion! Our verifier validates whether the final answer is correct. Here are two detailed examples of our data construction and verification process:
>
> For Sudoku, a typical board game, we identify the difficulty parameter as grid size and set it to 7 as example. Our generator creates a 7x7 grid instance with some missing numbers marked as X. We then format the instance with a prompt describing the Sudoku task to create the final question. Given the question, the LLM attempts to reason and solve the task, providing the final grid with filled values. Our verifier then checks whether the response's grid complies with Sudoku rules (such as single value rules) to verify correctness.
>
> For math path problems, an example instance is: $5-{?}-0 \times 9+{?} \times 5-3 \times ({?} \bmod {?}) \times 1-1 = 4$, where each ? represents a number to be solved. The identified difficulty parameter is the missing number ratio in the equation. The verifier validates whether the LLM's solution makes the equation work correctly.
>
> Therefore, our verifier checks answers’ correctness as in other RLVR works (rather than reasoning steps) and the verification is very accurate without apparent failing cases.
>
>
> >Q5: Including additional results on a distinct open-source model, such as LLaMA or Mistral, would strengthen the paper’s claim beyond Qwen-specific training dynamics.
>
> A5: Thank you for the suggestion! We conduct RL experiments on the mid-trained LLaMA-3.1-8B model [1] and observe significant improvements in both logical reasoning and math generalization performance. The results demonstrate that our SynLogic framework is effective across different model architectures, not limited to Qwen-specific training dynamics.
>
> | Model | SynLogic-val | KOR-Bench | BBH | BBEH | AIME 2024 | MATH 500 | AMC 2023 |
> |-------|-------------|-----------|-----|------|-----------|----------|----------|
> | LLaMA-3.1-8B (Base) | 4.2 | 37.2 | 43.5 | 4.1 | 3.3 | 38.7 | 20.3 |
> | LLaMA-3.1-8B (Trained on SynLogic) | 38.0 | 44.0 | 55.2 | 12.0 | 14.5 | 68.4 | 45.0 |
> | Improvement | +33.8 | +6.8 | +11.7 | +7.9 | +11.2 | +29.7 | +24.7 |
>
>
>
>
> >Q6: While the authors demonstrate meaningful generalization improvements over base models in math and coding tasks, it would be more informative to compare against the instruct-tuned versions (e.g., Qwen2.5-Instruct)
>
> >Q7: I would encourage the authors to include or discuss results where SynLogic data is used to fine-tune even stronger models
>
> A6 & A7: As suggested by the reviewer, we conduct RL training based on Qwen2.5-32B-Instruct, which is an instruct-tuned version and stronger than all the base models in our submission version. Results are shown below. Similarly, we observe consistent
> improvements on logical and math tasks, even though we only trained for 50 steps with 128 batch size due to time constraints.
>
> | Model | SynLogic-val | KOR-Bench | BBH | BBEH | AIME 2024 | MATH 500 | AMC 2023 |
> |-------|-------------|-----------|-----|------|-----------|----------|----------|
> | Qwen2.5-32B-Instruct| 12.0 | 54.7 | 84.5 | 17.5 | 10.0 | 82.2 | 57.5 |
> | Qwen2.5-32B-Instruct (Trained on SynLogic) | 31.7 | 58.1 | 85.7 | 20.3 | 16.25 | 81.8 | 63.4 |
> | Improvement | +19.7 | +3.4 | +1.2 | +2.8 | +6.25 | -0.4 | +5.9 |
>
> The improvements on logical benchmarks like KOR-Bench (+3.4) and math benchmarks like AIME 2024 (+6.25) validate the effectiveness of SynLogic on stronger, instruction version models.
>
>
>
> [1] Wang et al. OctoThinker: Mid-training Incentivizes Reinforcement Learning Scaling.  arXiv preprint. arXiv:2506.20512 2025

---

> > ### Comment · Reviewer_cYNo · 2025-08-04
> >
> > Thank you for the responses. I am impressed by the results that we can have further improvements on LLaMA and Qwen-Instruct models!!
> >
> > For Q3 & Q4, I strongly recommend you to introduce how to control the difficulties and how your verifiers work for EACH task.
> >
> > For Q2: How confident you are that after the training with your 10+ prompts, models can also work well on other prompts in these tasks? What is the prompt sensitivity of your proposed method?

---

> ### Author Response · Authors · 2025-08-05
>
> Thank you for your recognition of our further RL experiments!
>
> >Q: For Q3 & Q4, I strongly recommend you to introduce how to control the difficulties and how your verifiers work for EACH task.
>
> **A:** Thank you for the suggestion! We will surely provide the details on the difficulty control and verifier implementation in the next revision of the paper, as exemplified above in Q3 and Q4.
>
> >Q: For Q2: How confident are you that after training with your 10+ prompts, models can also work well on other prompts for these tasks? What is the prompt sensitivity of your proposed method?
>
> **A:** This is a good question. We are actually confident about the prompt generalization performance of our trained models. Below we provide two empirical evidence to support this:
>
>
> **Cross-Benchmark Validation:** Our significant improvements on external benchmarks like BBEH and KOR-Bench, which use prompt formats significantly different from our training data, demonstrate strong prompt generalization.
>
> **Comprehensive Prompt Robustness Test:** To directly address your concern about prompt sensitivity, we rely on Claude to generate 5 completely new prompt templates for each SynLogic task, with the minimum edit distance between new prompts and original prompts being 138.5 characters on average (average length of the prompts is 705 characters). We then evaluated our trained model across all these variants on SynLogic-Val dataset:
>
>
>
> | Models | SynLogic-Val Acc |
> |--------|------------------|
> | SynLogic-32B (w/ original prompt) | 52.9 |
> | w/ prompt v1 | 52.0 |
> | w/ prompt v2 | 51.4 |
> | w/ prompt v3 | 51.0 |
> | w/ prompt v4 | 53.8 |
> | w/ prompt v5 | 54.6 |
> | Average | 52.6 |
> | Standard Deviation | 1.4 |
>
>
> The results show  good consistency across different prompt formulations (mean: 52.6, std: 1.4), with performance even slightly improving over the original prompt in some cases. This demonstrates that our models have learned robust logical reasoning patterns rather than memorizing specific prompt formats, confirming robust prompt generalization  of our approach. We will include this prompt sensitivity analysis in the next revision of this paper.

---

> > ### Comment · Reviewer_cYNo · 2025-08-07
> >
> > Thanks for the further results. I have raised my score.

---

### Official Review · Reviewer_GS5X · 2025-06-28

**Clarity:** 2
**Significance:** 3
**Originality:** 3
**Rating:** 4
**Confidence:** 3

**Summary:**

This paper introduces SYNLOGIC, a novel framework and dataset designed to address the scarcity of diverse, high-quality, and verifiable logical reasoning data for training Large Language Models (LLMs). Specifically, the contributions are:
1. A Data Synthesis Framework: A systematic pipeline for generating logical reasoning problems across 35 diverse tasks (e.g., Sudoku, Cryptarithms, Arrow Maze). This framework allows for controlled synthesis with adjustable difficulty and includes a dedicated verifier for each task, making it ideal for RL with verifiable rewards (RLVR).
2. The SYNLOGIC Dataset: A large-scale dataset generated by the framework, with two versions tailored for different model sizes ("Easy" for 7B models and "Hard" for 32B models).
3. Empirical Validation and State-of-the-Art Results: The authors train Qwen2.5 models (7B and 32B) on SYNLOGIC using the GRPO reinforcement learning algorithm. Their models achieve state-of-the-art performance on logical reasoning benchmarks among open-source models, notably surpassing the strong DeepSeek-R1-Distill-Qwen-32B model by 6 points on BBEH.
4. Demonstration of Generalization and Efficiency: The paper shows that training on SYNLOGIC not only improves logical reasoning but also generalizes to mathematical tasks. Furthermore, experiments reveal that mixing SYNLOGIC data with math or code data significantly improves the training efficiency and final performance in those domains, highlighting the complementary nature of logical reasoning.

The authors commit to open-sourcing both the dataset and the synthesis pipeline to foster further research in general reasoning for LLMs.

**Questions:**

1. Is SYNLOGIC trained from qwen -base model or -instruct model in Table 2?
2. You may need to cite another paper [FineReason](https://arxiv.org/abs/2502.20238), which also releases the generation pipeline and data for logic data, and possibly compare the effectiveness of your data and this data.
3. Other questions in Weakness 1 and the difference between consumed math data and training steps mentioned in weakness 3.

**Ethical Concerns:**

["NO or VERY MINOR ethics concerns only"]

**Final Justification:**

The work is conducted on a large model size and improves the logic reasoning without hurting other capability. The authors say " The purpose of creating SynLogic is to provide researchers and practitioners with an RL logical reasoning dataset that can be used to enhance models’ logical reasoning performance through RL." Therefore, a good data source will be released, which is a good contribution.

**Limitations:**

yes

**Quality:**

3

**Strengths And Weaknesses:**

Strength:
1. Addresses a Clear and Important Problem: The paper correctly identifies a significant bottleneck in advancing general reasoning capabilities in LLMs—the lack of diverse, verifiable training data outside of math and code. Its focus on logical reasoning as a fundamental building block is a well-founded and timely hypothesis.
2. The authors commit to open-sourcing both the dataset and the synthesis pipeline.

Weakness:
1. The generation pipeline is too abstract and hard to understand. For example, in Line 102, it is difficult to understand how parameters for each task are used to control the synthesis process. In line 105, what is the output of your generator? Line 111, how do you perform automatic checks? Line 117, what are the chat models? Line 121, what is the conversion prompt? Line 124, Is the verification the same as that in in line 111?
2. The performance in Table 2 shows that Synlogic still lags behind R1-disill-qwen in 32B, limiting the effectiveness of Synlogic data.
3. For the experiments in Figure 5 and Figure 6, it seems the constructed logic data only do well on KOR-Bench, which can be the indomain nature of puzzle-related problems in KOR-bench. However, there is no significant effectiveness on math and code tasks.  Particularly, in Table 6-(b), using only code data has a higher effectiveness and greater training efficiency. Meanwhile, I also found it hard to understand the difference between consumed math data and training steps since the range of these two are the same in the figure.

---

> ### Author Rebuttal · Authors · 2025-07-31
>
> Thank you for the comments and we address them below.
>
> >Q1:  in Line 102, it is difficult to understand how parameters for each task are used to control the synthesis process.
>
> A1: Each logical reasoning task has specific parameters that control its difficulty level. For Sudoku, the key parameter is grid size (e.g., 5x5, 7x7, 9x9) - larger grids create more complex puzzles. For Cryptarithms, consider this example: SEND + MORE = MONEY, where each letter represents a unique digit, and this task is to find the numerical equation making the equation true. The difficulty parameters include the number of letters used and the length of arithmetic expressions. These parameters are systematically varied during synthesis to create problems of different difficulty levels. We apologize for incomplete details of this process in the submission and we will provide detailed parameter descriptions for all 35 tasks in the appendix in the next revision.
>
> >Q2: In line 105, what is the output of your generator? Line 111, how do you perform automatic checks?
>
> A2: The task generator outputs structured problem instances with specific formats. For Sudoku, it generates a grid with some numbers filled and others marked as "X" for missing values. For Cryptarithms, it outputs equations like "SEND + MORE = MONEY" where each letter represents a unique digit. The automatic solvability checks use algorithmic verification: Sudoku uses backtracking algorithms to verify solutions exist, while Cryptarithms uses constraint satisfaction to check for valid digit assignments. We will provide detailed output formats for all tasks in the appendix of the next version.
>
> >Q3:   Line 117, what are the chat models?
>
> A3: We use the Qwen-2.5-32B-Instruct model for this step.
>
> >Q4: Line 121, what is the conversion prompt?
>
> A4: The conversion prompt transforms structured problem instances into natural language instructions. For example, in Sudoku: "Here is a 7x7 Sudoku puzzle, where X represents a hidden number that needs to be filled in. Please solve it step by step." as demonstrated in Figure 1. For Cryptarithms, we use "Solve this cryptarithm: {equation}, where each letter represents a unique digit. Find the digit substitution that makes the equation true." We use multiple prompt templates for each task to ensure robust instruction following. We did not include all prompt templates in the submission because there are many of them, and we instead only included certain examples in Figure 1.
>
> >Q5: Line 124, Is the verification the same as that in line 111?
>
> A5: No, they are different steps in our pipeline. The automatic solvability checks (line 111) aim to filter out unsolvable problems before they enter the training dataset. In contrast, verification (line 124) happens during RL training to evaluate model responses and provide rewards. For example, the automatic solvability checks in Sudoku filter out created puzzles with no valid solutions using backtracking algorithms. The verification checks whether the model's final answer is correct (e.g., whether the filled Sudoku grid satisfies all rules).
>
> >Q6: The performance in Table 2 shows that Synlogic still lags behind R1-disill-qwen in 32B, limiting the effectiveness of Synlogic data.
>
> A6: We emphasize that Table 2 includes R1-disill-qwen for reference point only, and it is not fair to compare our RL models to R1-Distill-Qwen-32B due to fundamental differences in training approaches and resources. R1-Distill-Qwen-32B is trained on large-scale trajectory data from R1 models, which includes extensive supervised fine-tuning processes. In contrast, our SynLogic models are trained using only 33K synthetic logical reasoning instances through pure RL training without any additional supervised data.  The purpose of creating SynLogic is to provide researchers and practitioners with an RL logical reasoning dataset that can be used to enhance models’ logical reasoning performance through RL.
>
>
> >Q7: For the experiments in Figure 5 and Figure 6, it seems the constructed logic data only do well on KOR-Bench, which can be the indomain nature of puzzle-related problems in KOR-bench. However, there is no significant effectiveness on math and code tasks.
>
> A7: The key message from Figure 5 and Figure 6 is that our approach significantly enhances logical reasoning without hurting performance on other domains  under the same training compute -- that means including our dataset in the training recipe is like a free gift without negative transfer to other domains or consuming additional compute, as described in line 244-246. The effectiveness is particularly evident when comparing performance under equivalent math/code data consumption (Figures 5-c and 6-c), where mixed training shows clear advantages over training with math or code data alone.
>
>
> >Q8: Meanwhile, I also found it hard to understand the difference between consumed math data and training steps since the range of these two are the same in the figure.
>
> A8: We apologize for the confusion in Figure 5-c and Figure 6-c. The x-axis represents the number of consumed math/code data samples (step×batch_size), not training steps. We will update the axis labels in the next version. To clarify, when we mix logic data with math/code data, the total training steps remain the same, but we consume proportionally less math/code data since logic data occupies some of the training budget. For example, if we train for 1000 steps with pure math data, we consume 1000×batch_size math samples. But with mixed training (50% logic, 50% math), we still train for 1000 steps but only consume 500×batch_size math samples. Therefore, we plot Figures 5-c and 6-c to show performance under equivalent consumed math/code data. The key insight is that mixed training achieves better performance while using less domain-specific data, demonstrating the efficiency of our approach.
>
>
> >Q9: Is SYNLOGIC trained from qwen -base model or -instruct model in Table 2?
>
> A9: As described in line 167, we conduct RL training starting from Qwen-Base models.
>
> >Q10: You may need to cite another paper FineReason, which also releases the generation pipeline and data for logic data, and possibly compare the effectiveness of your data and this data.
>
> A10: Thank you for the suggestion, and we will surely cite it in the next revision. We also tried comparing it empirically during rebuttal. However, FineReason did not release their training data or the source code for their data generation pipeline. Below we describe the relation and distinction between SynLogic and FineReason.
>
> Both SynLogic and FineReason investigate logical reasoning tasks and conduct RL training experiments. However, there are significant differences between FineReason and SynLogic in both scope and methodology.
> 1. **Scale & Diversity**: SynLogic covers 35 distinct logical reasoning tasks (vs. FineReason's 4 puzzle tasks), providing much broader coverage of reasoning patterns and cognitive skills.
> 2. **Scalable Synthesis**: SynLogic features adjustable difficulty parameters and automated generation of unlimited training data with verifiable rewards, enabling efficient RL training. FineReason did not open-source their data generation code.
> 3. **Strong RL Scaling**: SynLogic conducts large and comprehensive RL experiments on these 35 logical reasoning tasks and math and coding domains over 7B and 32B models, while FineReason only conducts experiments on 4 logical tasks and math domain over smaller 1.5B and 7B models.

---

> > ### Comment · Reviewer_GS5X · 2025-08-06
> >
> > Your reponse mostly address my concerns, I will raise my score to accept this work.

---

### Official Review · Reviewer_wT7y · 2025-07-02

**Clarity:** 3
**Significance:** 2
**Originality:** 2
**Rating:** 4
**Confidence:** 3

**Summary:**

This paper studies the problems of reinforcement learning from verifiable feedback. The paper introduces a collection of synthetic logical reasoning tasks, called SynLogic, for training LLMs. SynLogic 35 36 logical reasoning tasks, where each of the tasks supports two difficulty levels and rule-based evaluators for reliable evaluation.

Using SynLogic, the paper trains Qwen-7B and 32B with reinforcement learning, which achieves strong performance on logical reasoning tasks in Big-bench-extra-hard, and also sees a bit of generalization to GPQA.

**Questions:**

See weakness for concerns.

**Ethical Concerns:**

["NO or VERY MINOR ethics concerns only"]

**Final Justification:**

I thank authors for the detailed response. I'd like to retain my score, slightly leaning the positive side. As in my response, I do appreciate the work for collecting this number of tasks and run experiments to study the benefits on RL on these tasks. At the same time, I feel 1) the main improvements on logical reasoning is somewhat in-domain (KOR bench, BBH, BBEH are mostly about games/logical puzzles) which is in some sense. similar to such collection of data. 2) on more OOD tasks like MATH, as in answers to Q3, the improvements are not consistent, 20 tasks < 10 tasks on MATH 500 and AMC 2023, 5 tasks work best on MATH 500.

**Limitations:**

yes

**Quality:**

3

**Strengths And Weaknesses:**

## Strength

The collected data suite, SynthLogic, contains over 30 logical reasoning tasks of multiple types. The tasks have some nice properties of having two difficulty levels and reliable rule-based evaluation.  This collection of data suites can be useful for the research community in the field of RLVR for reasoning.

The authors also conduct experiments of running RL over the logical reasoning problems, which shows good in-domain results and performance improvements on similar logical reasoning benchmarks (KOR and BBEH also have problems like logical, puzzle, cipher, which, in my opinion, are similar with the proposed data suite). The more interesting findings are including SynthLogic generalizes a bit to GPQA and LiveCodeBench.

The paper is mostly well-written and easy to follow.

## Weakness

The generalization to more realistic benchmarks (LiveCode and GPQA) shows observable but modest improvements considering the additional amount of training data.

While the paper compiles a set of tasks, most of them can be found in existing resources. The paper does not add new things for testing LLMs.

The paper presents a collection of logical reasoning models and empirical study on using them for training models. I feel the paper could benefit from a bit more analysis, such as impact of additional training data size, generalization across tasks within SynLogic.

The experiment only contains one model family (Qwen). It is unsure how the results could generalize across different model families like Llama.

---

> ### Author Rebuttal · Authors · 2025-07-31
>
> Thanks for your helpful comments; we are glad that you acknowledged our work's strengths! Below we address your concerns:
>
> >Q1: The generalization to more realistic benchmarks (LiveCode and GPQA) shows observable but modest improvements considering the additional amount of training data.
>
> A1: Thank you for the comment. We first note that, in all table comparisons, different RL training runs are always benchmarked using the same number of training steps. Therefore, our model does not consume more training tokens than the baselines, even when additional datasets are incorporated. More importantly, we emphasize that the contribution of SynLogic, as shown in the tables, is that it **greatly improves logical reasoning performance without negatively impacting other domains, all within the same training compute budget**. This means that including our dataset in the training pipeline acts as a free addition—providing significant benefits on logical reasoning without causing negative transfer or requiring extra computation. Consequently, the modest improvements observed on LiveCode and GPQA do not undermine our main contribution or the claims as improving them is not the main goal.
>
> >Q2: While the paper compiles a set of tasks, most of them can be found in existing resources. The paper does not add new things for testing LLMs.
>
> A2: As discussed in Section 2.2 (The Data Synthesis Framework), although most tasks can be found in existing resources, they do not open-source their synthesis code and do not conduct careful difficulty control. We have overcome this challenge through the time-consuming development of these tasks’ generation framework. Furthermore, our work focuses on conducting RLVR on diverse and scalable logical tasks rather than introducing new testing paradigms for LLMs.
>
> >Q3:  I feel the paper could benefit from a bit more analysis, such as impact of additional training data size, generalization across tasks within SynLogic.
>
> A3: Thanks for your suggestions! Here, we conduct ablation studies on different data sizes by controlling the selected task numbers. Using the Qwen2.5-7B-Base model, we conduct RL training varying the logical reasoning tasks: 10 tasks (5K samples), 20 tasks (10K samples), and all 27 tasks (16K samples) under the same training steps. The results are shown below:
>
> | Tasks | SynLogic-val | KOR-Bench | BBH | BBEH | AIME 2024 | MATH 500 | AMC 2023 |
> |-------|-------------|-----------|-----|------|-----------|----------|----------|
> | 10 tasks (5K) | 32.6 | 44.2 | 66.0 | 5.6 | 10.0 | 72.1 | 52.5 |
> | 20 tasks (10K) | 33.4 | 47.7 | 66.0 | 7.3 | 11.25 | 70.1 | 47.1 |
> | All tasks (16K) | **44.4** | **48.1** | **66.5** | **8.0** | 10.0 | 71.8 | **55.0** |
>
> The results show that increasing the number of tasks generally improves performance across most benchmarks, demonstrating the effectiveness of our diverse task collection.
>
>
>
>
> >Q4: The experiment only contains one model family (Qwen). It is unsure how the results could generalize across different model families like Llama.
>
>
> A4: Thank you for the suggestion! We conduct RL experiments on the mid-trained LLaMA-3.1-8B model [1] and observe significant improvements in both logical reasoning and math generalization performance. The results demonstrate that our SynLogic framework is effective across different model architectures, not limited to Qwen-specific training dynamics.
>
> | Model | SynLogic-val | KOR-Bench | BBH | BBEH | AIME 2024 | MATH 500 | AMC 2023 |
> |-------|-------------|-----------|-----|------|-----------|----------|----------|
> | LLaMA-3.1-8B (Base) | 4.2 | 37.2 | 43.45 | 4.1 | 3.3 | 38.7 | 20.3 |
> | LLaMA-3.1-8B (Trained on SynLogic) | 38.0 | 44.0 | 55.2 | 12.0 | 14.5 | 68.4 | 45.0 |
> | Improvement | +33.8 | +6.8 | +11.75 | +7.9 | +11.2 | +29.7 | +24.7 |
>
> [1] Wang et al. OctoThinker: Mid-training Incentivizes Reinforcement Learning Scaling. arXiv preprint. arXiv:2506.20512 2025

---

> > ### Comment · Reviewer_wT7y · 2025-08-05
> >
> > Thank you for your response. I'd like to retain my score, slightly leaning the positive side. As in my response, I do appreciate the work for collecting this number of tasks and run experiments to study the benefits on RL on these tasks. At the same time, I feel 1) the main improvements on logical reasoning is somewhat in-domain (KOR bench, BBH, BBEH are mostly about games/logical puzzles) which is in some sense. similar to such collection of data. 2) on more OOD tasks like MATH, as in answers to Q3, the improvements are not consistent, 20 tasks < 10 tasks on MATH 500 and AMC 2023, 5 tasks work best on MATH 500.

---

### Official Review · Reviewer_7q1L · 2025-07-05

**Clarity:** 4
**Significance:** 4
**Originality:** 4
**Rating:** 5
**Confidence:** 4

**Summary:**

The paper introduces SYNLOGIC, a comprehensive data synthesis framework and dataset designed to generate diverse logical reasoning tasks at scale. SYNLOGIC addresses the gap in high-quality logical reasoning training data, which is essential for developing general reasoning capabilities in large language models (LLMs). The dataset includes 35 diverse tasks, each with task-specific generation code, a verification rule, and configurable difficulty parameters. This allows for the creation of data with controlled difficulty levels, making it suitable for reinforcement learning (RL) with verifiable rewards.

**Questions:**

- Consider implementing a dynamic difficulty adjustment mechanism where the difficulty of the tasks is gradually increased during training. This could involve starting with simpler tasks and gradually introducing more complex ones as the model's performance improves.
- Conduct additional experiments to compare the generalization performance of the 7B and 32B models when trained with mixed SYNLOGIC, mathematical, and coding data. This could include a more granular analysis of performance on specific benchmarks and tasks.
- Perform ablation studies to investigate the impact of dataset size on training efficiency and performance. This could involve training models with different sizes of SYNLOGIC data (e.g., 10k, 20k, 30k, 33k samples) and comparing the results.
- Conduct experiments to investigate the integration of SYNLOGIC with other training techniques. This could include few-shot learning, fine-tuning on specific tasks, or hybrid approaches that combine multiple techniques.

**Ethical Concerns:**

["NO or VERY MINOR ethics concerns only"]

**Limitations:**

yes

**Quality:**

3

**Strengths And Weaknesses:**

Strengths:
The paper introduces SYNLOGIC, a comprehensive dataset containing 35 diverse logical reasoning tasks. This is a significant contribution to the field, as it addresses the lack of high-quality logical reasoning training data.
The framework allows for controlled synthesis of data with adjustable difficulty and quantity, which is crucial for training models effectively.
All examples in the dataset can be verified by simple rules, making them ideally suited for reinforcement learning (RL) with verifiable rewards.
The authors validate the effectiveness of RL training on the SYNLOGIC dataset using 7B and 32B models, demonstrating state-of-the-art performance on logical reasoning benchmarks.
he mixed training approach, combining SYNLOGIC data with mathematical and coding tasks, improves training efficiency and significantly enhances reasoning generalization across multiple domains.
The authors provide a detailed description of the SYNLOGIC framework, including task selection, parameter identification, logic instance generation, and verification suite. This makes the work reproducible and transparent.

Weaknesses:
Due to limited computational resources, the authors do not perform precise difficulty tuning for every task. This could mean that the data is not fully optimized for training 7B and 32B models.
The paper does not implement training with dynamically adjustable difficulty, where sample complexity progressively increases during training. This could limit the potential for further enhancing logical reasoning capabilities.

---

> ### Author Rebuttal · Authors · 2025-07-31
>
> We sincerely appreciate your thoughtful comments and recognition of the strengths of our work. Below, we address each of your concerns and questions in detail:
>
>
>
> >Q1: Conduct additional experiments to compare the generalization performance of the 7B and 32B models when trained with mixed SYNLOGIC, mathematical, and coding data. This could include a more granular analysis of performance on specific benchmarks and tasks.
>
> A1:  To provide more granular analysis, here we analyze the effect of mixed training on response length and training efficiency beyond accuracy on AIME 2024.  Specifically, based on the 7B model trained on mixed synlogic, math, and code data, we compare its performance to training on math-only data. Results on AIME 2024 are shown below. We find that mixed training leads to higher performance under the same training steps (while consuming less math data).  Also, mixed training leads to longer responses.
>
> | Training Steps | 48 | 96 | 144 | 192 | 240 | 304 |
> |----------------|----|----|----|----|----|----|
> | Response Length (mix-logic vs. math-only) | **1122**/1160 | **1366**/1312 | **1640**/1554 | **2716**/1915 | **3926**/3813 | **8621**/7260 |
> | Performance (mix-logic vs. math-only) | **11.6**/11.6 | **12.5**/12.9 | **15.8**/14.6 | **16.7**/15.4 | **17.1**/16.7 | **18.3**/17.1 |
>
>
>
> >Q2: Perform ablation studies to investigate the impact of dataset size on training efficiency and performance.
>
>
> A2: Thank you for the suggestion.  Here, we conduct ablation studies on different data sizes by controlling the selected task numbers. Using the Qwen2.5-7B-Base model, we conduct RL training varying the logical reasoning tasks: 10 tasks (5K samples), 20 tasks (10K samples), and all 27 tasks (16K samples) under the same training steps. The results are shown below:
>
> | Tasks | SynLogic-val | KOR-Bench | BBH | BBEH | AIME 2024 | MATH 500 | AMC 2023 |
> |-------|-------------|-----------|-----|------|-----------|----------|----------|
> | 10 tasks (5K) | 32.6 | 44.2 | 66.0 | 5.6 | 10.0 | 72.1 | 52.5 |
> | 20 tasks (10K) | 33.4 | 47.7 | 66.0 | 7.3 | 11.25 | 70.1 | 47.1 |
> | All tasks (16K) | **44.4** | **48.1** | **66.5** | **8.0** | 10.0 | 71.8 | **55.0** |
>
> The results show that increasing the number of tasks generally improves performance across most benchmarks, demonstrating the effectiveness of our diverse task collection.
>
>
> >Q3: Conduct experiments to investigate the integration of SYNLOGIC with other training techniques.
>
> A3: Thanks for your suggestion.  Our paper primarily focuses on the RLVR setting, which we believe is sufficiently substantive to stand on its own. We look forward to investigating the impact of SYNLOGIC in other training paradigms like SFT in future work.
>
>
> >Q4: Consider implementing a dynamic difficulty adjustment mechanism where the difficulty of the tasks is gradually increased during training.
>
> A4: Thank you for your suggestions. Curriculum learning as advised here may indeed help learn logical reasoning greatly. Due to time and resource limitations we were unable to conduct this experiment during rebuttal, but we will consider exploring it in future work.

---

> > ### Comment · Reviewer_7q1L · 2025-08-06
> >
> > Thank authors for detailed response. The added experiments (e.g., mixed training analysis, ablation studies) strengthen the paper’s claims and highlight the dataset’s versatility. It is somewhat regrettable that the paper did not try curriculum learning-based training strategies, which might have led to even better results.

---

### Decision · Program_Chairs · 2025-09-17

**Decision:**

Accept (poster)

**Comment:**

The paper introduces a synthetic benchmark combining many logic problems, finetunes LMs with RL on it, and show some cross-domain generalization to benchmarks such as LiveCodeBench.

All reviewers agreed that the combination of the benchmark and RL training experiments make it a worthy contribution. A weakness is that cross-domain generalization is not strong and consistent across benchmarks, but paper contains an interesting idea that can be explored further. I recommend the authors to include the rebuttal extensions in the updated paper.